# The Role of Type 2 Diabetes in Patient Symptom Attribution, Help-Seeking, and Attitudes to Investigations for Colorectal Cancer Symptoms: An Online Vignette Study

**DOI:** 10.3390/cancers15061668

**Published:** 2023-03-08

**Authors:** Lauren Smith, Christian Von Wagner, Aradhna Kaushal, Meena Rafiq, Georgios Lyratzopoulos, Cristina Renzi

**Affiliations:** 1Research Department of Behavioural Science and Health, University College London, London WC1E 6BT, UK; laurenmargaretsmith@ymail.com (L.S.);; 2Centre for Cancer Research and Department of General Practice, University of Melbourne, Melbourne 3052, Australia; 3Faculty of Medicine, University Vita-Salute San Raffaele, 20132 Milan, Italy

**Keywords:** colorectal cancer, diabetes, cancer diagnosis

## Abstract

**Simple Summary:**

Diabetic individuals have lower cancer awareness and are two-fold more likely than non-diabetics to attribute some red-flag cancer symptoms to medications.

**Abstract:**

Objectives: Type 2 diabetes is associated with a higher risk of colorectal cancer (CRC) and advanced-stage cancer diagnosis. To help diagnose cancer earlier, this study aimed at examining whether diabetes might influence patient symptom attribution, help-seeking, and willingness to undergo investigations for possible CRC symptoms. Methods: A total of 1307 adults (340 with and 967 without diabetes) completed an online vignette survey. Participants were presented with vignettes describing new-onset red-flag CRC symptoms (rectal bleeding or a change in bowel habits), with or without additional symptoms of diabetic neuropathy. Following the vignettes, participants were asked questions on symptom attribution, intended help-seeking, and attitudes to investigations. Results: Diabetes was associated with greater than two-fold higher odds of attributing changes in bowel habits to medications (OR = 2.48; 95% Cl 1.32–4.66) and of prioritising diabetes-related symptoms over the change in bowel habits during medical encounters. Cancer was rarely mentioned as a possible explanation for the change in bowel habits, especially among diabetic participants (10% among diabetics versus 16% in nondiabetics; OR = 0.55; 95% CI 0.36–0.85). Among patients with diabetes, those not attending annual check-ups were less likely to seek help for red-flag cancer symptoms (OR = 0.23; 95% Cl 0.10–0.50). Conclusions: Awareness of possible cancer symptoms was low overall. Patients with diabetes could benefit from targeted awareness campaigns emphasising the importance of discussing new symptoms such as changes in bowel habits with their doctor. Specific attention is warranted for individuals not regularly attending healthcare despite their chronic morbidity.

## 1. Introduction

Colorectal cancer (CRC) is the fourth most common cancer in the UK, with the second highest cancer-related mortality. Currently, large proportions of CRCs are diagnosed at an advanced stage (52%) or following emergency presentation (24%) in the UK [1]. A recent international study found that 23% to 36% of CRCs are diagnosed as an emergency [2]. Diagnosing cancer earlier is a key public health target for improving survival [3,4].

Diabetes is associated with an increased risk of developing CRC [5] through complex biological mechanisms related to insulin-like growth factors, insulin resistance, compensatory increased insulin levels, and prolonged hyperglycaemia [6,7]. Moreover, individuals with diabetes have a higher risk of being diagnosed with CRC at an advanced stage [8] and have worse outcomes than nondiabetics [9,10,11,12].

The time it takes for patients to appraise symptoms and seek help is a significant contributor to the overall delay before a cancer is diagnosed [13], but research on how chronic conditions impact this delay is scant [14,15]. When experiencing potential CRC symptoms such as a change in bowel habits, individuals with diabetes may consider alternative explanations, attributing the symptoms to their pre-existing condition or to medications [16,17]. Diabetic neuropathy can cause changes in bowel habits, as can diabetic medications such as metformin, whose possible side effects include constipation and diarrhoea [12]. Competing demands might also contribute to diagnostic delays, especially when the management of diabetes is complex and is prioritised by patients and doctors over the investigation of new symptoms of an as-yet-undiagnosed cancer. It has also been hypothesised that diabetes may facilitate, rather than hinder, timely cancer diagnosis [18]. This is because diabetes, like other chronic conditions, might be associated with frequent healthcare contacts [17,18], leading to opportunities to discuss new symptoms and possibly reducing delays in CRC diagnosis through what is termed a surveillance mechanism or surveillance effect [19].

The aim of the study was to investigate symptom attribution, intended help-seeking, and willingness to undergo investigations for potential CRC symptoms among people living with type 2 diabetes (hereafter, diabetes) compared with individuals without diabetes.

## 2. Methods

We performed an online cross-sectional vignette survey asking participants about the action they would take after reading a vignette describing symptoms such as rectal bleeding or change in bowel habits, with or without additional symptoms of diabetic neuropathy in the feet. The word “cancer” was not mentioned to the study participants in order to mask the study aim and to reduce priming and response bias, similar to previous studies [14].

Vignettes are short, hypothetical scenarios echoing real-life situations [20,21]. They allow the manipulation of symptoms whilst keeping the context and environment constant to explore reactions and intended behaviours [22]. Vignettes have often been used in diagnostic research [23,24] and are particularly useful when investigating complex phenomena such as comorbidity-specific effects [14,23].

The study was approved by UCL Ethics committee (N14687/006).

### 2.1. Study Participants

Participants were recruited in August 2021 through Prolific, a survey provider (www.prolific.co, accessed on 28 February 2023). Eligibility criteria included age 50 years or above, resident in the UK, and no cancer diagnosis in the last 5 years. The age of 50+ was chosen due to the increasing prevalence with older age of both type 2 diabetes [25] and CRC [26]. At the time of the study, Prolific had ~5500 participants in the UK aged over 50, of which 62% were aged 50–59 and 64% were female. Participants meeting the eligibility criteria were contacted by email and asked if they would be interested in participating in a study on symptom perception and help-seeking. Participants were compensated £1.25 for completion of the survey, which is based on the questionnaire taking approximately 15 min to complete and a £5 per hour payment. This amount was the standard set by Prolific, who recruited participants from their panel, and is aimed at compensating participants for the time taken to complete the survey.

Quota sampling was used to try to ensure that 50% of the sample were people with diabetes. To facilitate the recruitment of a sufficiently large number of diabetic participants the survey was additionally circulated to a local diabetes group. A total sample of *n* = 2000 participants was estimated to provide 80% power to detect a difference of 10% in anticipated help-seeking between people with and without diabetes at a significance level of *p* < 0.05.

### 2.2. Vignettes

We developed three vignettes, all of which described symptoms that could be indicative of CRC, and one of which additionally included symptoms of diabetic neuropathy.

Vignette 1 focused on rectal bleeding: When you use the bathroom, you notice blood in your poo (rectal bleeding). Other than this symptom you have noticed no other changes.

Vignette 2 focused on a change in bowel habits: You notice you have had changes in your normal bowel habit (such as looser poo, pooing more often or constipation). Other than this symptom you have noticed no other changes.

Vignette 3 focused on the co-occurrence of diabetic neuropathy and CRC symptoms: You notice you have numbness, tingling and pain in your feet, along with changes in your normal bowel habits (such as looser poo, pooing more often or constipation) and blood in your poo (rectal bleeding). Other than these symptoms you have noticed no other changes.

Changes in bowel habits and rectal bleeding are red-flag CRC symptoms warranting an urgent investigation via the suspected cancer pathway, according to UK NICE guidelines [27]. Symptoms indicating neuropathy in the feet were chosen because these could be concerning to the individual but are different from CRC symptoms, allowing us to examine variations in prioritisation by patients during medical encounters when experiencing multiple symptoms.

The vignettes and study material were co-designed with contributions from patient representatives, clinicians, and researchers. Twenty-two cognitive interviews and a pilot study with 200 participants were performed to ensure the study material was patient-centred and easy to understand [28]. Based on the feedback received, minor changes were made; for example, giving a clearer explanation of some words (e.g., endocrinologist).

Participants were randomly assigned to read either vignette 1 and 2 or vignette 3.

Following the vignettes, participants were asked precoded and open questions on symptom attribution, intended help-seeking, and attitudes to investigations (Appendix A). Additionally, participants provided information on their age, sex, ethnicity, educational level, and self-reported history of chronic conditions, using a question and precoded list of conditions adapted from the GP Patient Health Survey [29]. Answers to this question were used to classify participants into those with or without a pre-existing diagnosis of type 2 diabetes. Respondents with diabetes were presented with additional questions on their diabetes management.

### 2.3. Outcome Variables

#### 2.3.1. Symptom Attribution

Symptom attribution was explored using free-text responses, inviting participants to write down anything they thought may be a possible cause of the symptoms. Similar to previous research [14], we used content analysis to code answers as referring to cancer, benign gastrointestinal (GI) conditions, diabetes, or other conditions.

#### 2.3.2. Intended Help-Seeking

Intended help-seeking was measured by asking what action people would take. Precoded answers such as “Talk to members of your family” and “Contact the GP” were presented (full list in Appendix A), with an additional free-text option. Precoded answers were randomised to avoid order effects. In the analysis stage, answers of “probably would” and “definitely would” were combined into a “would take action” category.

#### 2.3.3. Willingness/Attitudes to Undergo Diagnostic Investigations

Participants were asked if they would be willing to have a colonoscopy/sigmoidoscopy and/or stool test (using yes/no response options) following the symptomatic presentation. If participants answered “no,” they were asked to clarify why.

#### 2.3.4. Prioritisation of Symptoms

Participants randomised to vignette 3 (co-occurrence of diabetic neuropathy and CRC symptoms) were asked to provide a ranking of the order in which they would mention each symptom to their GP.

### 2.4. Main Explanatory Variables

Having a diagnosis of type 2 diabetes was the main explanatory variable considered. In addition, those with diabetes were asked how often their diabetes is reviewed, with precoded answers of “at least once per year by the GP or by the specialist” or “not being regularly checked by a doctor or nurse”. They were asked what their HbA1c is (either mmol or %) and how they would describe their management of diabetes (“very good”, “good”, “average”, “bad”, or “very bad”; subsequently recoded as “good”, “average”, or “bad”. Based on clinical cut-offs, a HbA1c of 48 mmol/mol indicated a blood sugar level in the diabetic range; 48 and below indicated well controlled diabetes.

Additionally, past faecal occult blood test or colonoscopy/sigmoidoscopy information was collected by asking participants: “Have you ever had a stool sample?” and “Have you ever had a colonoscopy/sigmoidoscopy?”, with pre-coded answers: “no”, “yes, for screening”, and “yes, for symptoms”. 

### 2.5. Statistical Analysis

Chi-squared tests were used to compare the characteristics of participants with versus without diabetes. Multivariable logistic regression was used to explore the associations between diabetes and the following outcomes: symptom attribution, intended help-seeking, and willingness to undergo diagnostic investigations. Each outcome was examined in a separate multivariable model, adjusting for variables considered a priori as potential confounders based on the previous literature and on clinical reasoning (including age, sex, ethnicity, previous diagnostic testing, and total number of chronic morbidities).

Additionally, for the subgroup with diabetes, we evaluated the association between diabetes management and intended help-seeking for possible CRC symptoms, combining the participants across all three vignettes.

## 3. Results

### 3.1. Sample Characteristics

A total of 1456 participants agreed to take part in the study. Excluding 108 with incomplete responses and 59 with cancer in the last 5 years, a total of 1287 participants were included. Among respondents, 320 had diabetes and 967 were nondiabetics, 60.8% of the sample were female, and 87.3% were from a white ethnic background (Table 1), which is in line with Prolific’s participant characteristics. Diabetic respondents were older and more frequently had 3+ chronic conditions and a history of faecal occult blood test or colonoscopy/sigmoidoscopy (for screening or symptoms). Within the subgroup with diabetes, 16.6% had HbA1c results <48 mmol, 26.3% ≥48 mmol, and 57.2% unknown. Most self-rated their diabetes management as “good” (56.9%), with 10.6% self-rating it as “bad,” and 10.9% did not have their diabetes checked at least once a year (in either primary or secondary care).

### 3.2. Symptom Attribution

The change in bowel habits (vignette 2) was most frequently attributed to dietary changes by both the diabetic and nondiabetic participants, without a significant difference (30.9% vs. 34.2%, respectively). Whilst cancer was the second most frequent symptom attribution overall, participants with diabetes, compared with those without, were less likely to attribute the change in bowel habits to cancer (9.7%, vs. 15.9%; adjusted OR: 0.55, 95% CI: 0.36–0.85). Diabetic participants were also less likely to attribute the change in bowel habits to other bowel diseases (8.1% vs. 12.5 %; adjusted OR: 0.57, 95% Cl: 0.35–0.91) and to more often think that this symptom could be due to medications (7.5% vs. 2.5%; OR: 2.48, 95% Cl: 1.32–4.66) (Figure 1).

No significant differences between participants with and without diabetes were found in the case of rectal bleeding (vignette 1), with the most frequently mentioned symptom attributions being haemorrhoids (29.7%) and cancer (29.3%) (details in Appendix A). Multivariable logistic regression odds ratios for symptom attribution for vignettes 1 and 3 are reported in Appendix A.

### 3.3. Intended Help-Seeking

The most frequently reported action in case the participants were to experience a change in bowel habits (vignette 2) was “wait and see what happens”, irrespective of diabetes status (88% in both diabetic and nondiabetic participants). Among participants with diabetes, this was followed by mentioning the symptom to the GP if being seen for something else (78.5% with diabetes, 70.9% participants without diabetes).

In the case of rectal bleeding (vignette 1) or co-occurrence of CRC symptoms with numbness/pain in the feet (vignette 3), 91% of participants with diabetes would mention these symptoms to the GP if seeing them for something else. Among participants without diabetes the majority would look up these symptoms online (88.4% for vignette 1, 90.8% for vignette 3).

Compared with those without diabetes, participants with diabetes were more likely to contact the GP to seek help when experiencing rectal bleeding (41% versus 18%, adjusted OR: 1.62, 95% CI: 1.02–2.55) or a change in bowel habits (27% versus 7% adjusted OR: 1.70, 95% CI: 1.13–2.57) (Table 2). Diabetes was not associated with help-seeking when potential cancer symptoms co-occurred with numbness/pain in the feet (vignette 3), (adjusted OR: 1.28 95% CI: 0.72–2.28).

In all scenarios, an awareness that the symptom might be linked to cancer was associated with an increased likelihood of seeking help from a GP (rectal bleeding OR: 2.52, 95% CI: 1.76–3.61; change in bowel habits OR: 2.79, 95% CI: 1.92–4.04; co-occurrence of rectal bleeding, change in bowel habits, and numbness/tingling in the feet: OR: 3.90 95% CI: 2.43–6.25).

### 3.4. Diabetes Condition Management

Among the subgroup of participants with diabetes, we examined the likelihood of seeking help from a GP or a nurse for any of the CRC symptoms described in the vignettes, by self-reported diabetes management. Not having an annual check-up reduced the likelihood of help-seeking (adjusted OR: 0.23; 95% CI: 0.10–0.50), as did self-perceived poor diabetes control (adjusted OR: 0.20; 95% CI: 0.86–0.47) (Table 3). Higher than recommended HbA1c increased the odds of seeking help (adjusted OR: 3.7; 95% CI: 1.36–10.50). 

### 3.5. Willingness/Attitudes to Undergo Diagnostic Investigations for Symptoms

Among the total study sample, 98.4% were willing to take a stool test, and 95.6% were willing to have a colonoscopy in case they experienced rectal bleeding or a change in bowel habits, with no differences between those with or without diabetes.

At multivariable analysis there was no significant difference in the likelihood of taking part in diagnostic testing by diabetes status, adjusting for a history of stool test or colonoscopy/sigmoidoscopy, age, sex, ethnicity, and comorbidity number. Having a history of stool test (OR: 0.16; 95% CI: 0.04–0.35) or colonoscopy (OR: 0.28; 95% CI: 0.12–0.63) decreased the likelihood of being willing to have a test in the case of symptoms, compared with never having been tested. Men were less likely to be willing to have a colonoscopy compared with women (OR: 0.37; 95% CI: 0.19–0.72) (details in Appendix A).

In those unwilling to undergo investigations, the predominant reason was embarrassment in having a stool test (19%) or a colonoscopy (26.7%) (further details in Appendix A).

### 3.6. Patient Prioritisation of Symptoms When Communicating with the GP

When examining symptoms mentioned as the priority during the GP consultation, significant differences were found by diabetes status: a lower proportion of the diabetic participants mentioned rectal bleeding as the first priority compared with the nondiabetic participants (65.6% versus 77.0%, *p* = 0.004), whilst a higher proportion of the diabetic participants prioritised numbness/pain in the feet (24.2% versus 13.3%, *p* = 0.001). Change in bowel habits was considered a priority by a minority among both the diabetic and nondiabetic participants (9.9% versus 10.3%, *p* = 0.871). (Full result in Appendix A)

## 4. Discussion

### 4.1. Main Findings and Comparison with the Literature

Individuals with diabetes compared with those without were less likely to attribute possible CRC symptoms, such as a change in bowel habits, to cancer and more likely to think it might be caused by medications. When seeing their doctor, diabetic individuals were also more likely to prioritise concerns related to their chronic condition rather than discuss typical red-flag CRC symptoms such as a new-onset change in bowel habits. Diabetic individuals not regularly attending healthcare were less likely to seek help if experiencing red-flag cancer symptoms. 

Participants did not provide details on specific medicines they considered possibly linked to the change in bowel habits. However, metformin is a cornerstone of diabetes treatment [30,31], and it can lead to changes in bowel habits as a common side effect [12]. This might explain the patients’ interpretation of the symptom as being due to an inconsequential cause rather than cancer, in line with the “alternative explanation” hypothesis. Nonrecognition of cancer alarm symptoms can delay help-seeking [19,32], and comorbidities providing plausible “alternative explanations” have been previously associated with advanced-stage cancer diagnosis [33]. 

In contrast, having a chronic condition might also lead to more frequent healthcare contacts and opportunities to report possible cancer symptoms to the doctor, in line with the surveillance mechanism [19]. This is supported by the present study, indicating that compared with those without, individuals with diabetes would more likely seek help from a GP, and they would mention a change in bowel habits when attending for other reasons. However, the study shows that this was not the case when cancer symptoms co-occur with diabetic complications. In these circumstances, patients prioritised the symptoms of diabetic neuropathy over the cancer alarm symptoms when communicating with the doctor. This supports the competing demands theory [34] and might at least partially explain why some patients with chronic conditions might have a higher risk of diagnostic delays or advanced stage or emergency cancer diagnosis [33,35] despite increased GP consultations.

Diabetic patients with no annual checks or poor self-management who may have lower levels of primary care use were also found to be significantly less likely to seek help in the current study. This is in line with some previous evidence of an association between poorly controlled diabetes and late-stage cancer [8,36]. 

Whilst a high percentage of participants were willing to have stool tests and colonoscopies, those who had a history of previous testing were less willing to take part again, even if experiencing red-flag symptoms such as rectal bleeding or changes in bowel habits. Past research found that an all-clear result might lead to a false sense of security and over-reassurance, which can subsequently decrease the likelihood of help-seeking or prompt investigations if the same alarm symptom were to reoccur [37].This may possibly lead to missed opportunities for a prompt diagnosis.

### 4.2. Implications for Research and Practice

Despite being at an increased risk of developing CRC, people with diabetes have a lower likelihood of attributing typical CRC alarm symptoms to cancer and of prioritizing communication of cancer symptoms during a medical consultation. It is important for cancer awareness campaigns [38] to target people with common chronic conditions associated with a higher risk of cancer, promoting earlier symptom recognition and appropriate communication of new symptoms to the doctor. Direct questions and symptom elicitation focusing on red-flag symptoms during medical visits performed for diabetes management could also be useful, possibly adding key CRC screening questions to the Quality and Outcomes Frameworks (QOF). This might ensure it is proactively elicited at annual checks regardless of patient prioritisation. The study also stresses the importance of dedicating specific attention to patients with diabetes who are not likely to attend health care regularly and/or are likely have poor glycaemic control. Additional QOF incentives could be offered for practices able to engage with patients who previously missed diabetic reviews. Patient awareness and understanding of the common side effects of medication such as metformin could also be increased, as well as emphasising the importance of help-seeking if patients experience persistent changes in bowel habits. As part of the medication reviews of patients on metformin, questions regarding possible changes of bowel habits could be introduced. 

A recent study reported significantly longer intervals from first symptomatic presentation in primary care to investigations and cancer diagnosis for patients with pre-existing conditions such as diabetes [35]. Future research could explore factors contributing to longer primary care and diagnostic intervals, including whether clinical decision making on referrals for patients presenting with possible cancer symptoms varies by diabetes status. 

### 4.3. Strengths and Limitations

The study used online vignettes, a methodology which has an established record in diagnostic research for elucidating the cognitive and attitudinal drivers of behaviour. The methodology allows for the manipulation of conditions in standardised ways and has been shown to be an important methodological tool when it would be impossible to manipulate symptoms, conditions, and comorbidities to investigate the topics using other methods [39,40]. The inclusion of open-ended questions allowed for better understanding of the participant perspective, gaining insights into how they would communicate sensitive issues to their GP. A further strength is a high survey completion (88%).

An inherent limitation of vignette studies is that they examine intended rather than actual behaviour. The symptoms are simulated, and pain, discomfort, and worry about symptoms are not necessarily experienced in the same way as in real life. Whilst intention to act does not automatically equate to a behaviour actually occurring, expressing intentions is an important preliminary step in producing the behaviour. 

## 5. Conclusions

The study found that compared with those without, people with diabetes are less likely to attribute red-flag CRC symptoms such as changes in bowel habits to cancer, and they are more likely to attribute them to medications. Individuals with diabetes are also less likely to prioritise the reporting of possible new-onset cancer symptoms over diabetes-related symptoms during medical encounters. Interventions are needed encouraging healthcare visits for individuals not regularly attending healthcare despite their chronic morbidity. The findings can inform cancer awareness campaigns and clinical guidelines, targeting individuals with common conditions such as diabetes who are at higher risk of CRC, in order to help diagnose cancer earlier.

## Figures and Tables

**Figure 1 cancers-15-01668-f001:**
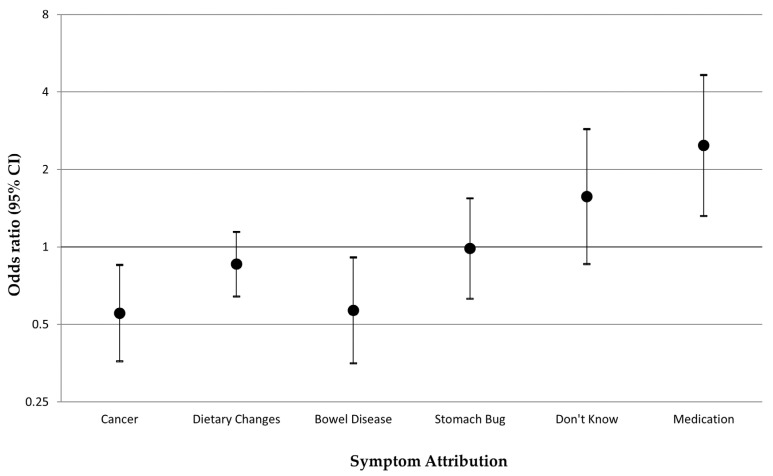
Symptom attributions reported by participants with diabetes versus those without when experiencing a change in bowel habits (vignette 2): multivariable logistic regression odds ratios, adjusted for age, sex, ethnicity, and comorbidity number. Note: Each symptom attribution was included as the outcome (binary yes/no outcome) in a separate multivariable model.

**Table 1 cancers-15-01668-t001:** Participant Characteristics.

	Total N = 1287	With Diabetes N = 320	Without Diabetes N = 967	*p*-Value (χ^2^ Test)
**Age (years)**				<0.001
50–59	791 (61.5%)	158 (49.4%)	633 (65.5%)	
60–69	399 (31.0%)	121 (37.8%)	278 (28.7%)	
70+	97 (7.5%)	41 (12.8%)	56 (5.8%)	
**Sex**				<0.001
Male	500 (38.9%)	158 (49.4%)	342 (35.4%)	
Female	782 (60.8%)	161 (50.3%)	621 (64.2%)	
Prefer not to say #	5 (0.4%)	1 (0.3%)	4 (0.4%)	
**Ethnic Group**				0.037
White British	1123 (87.3%)	290 (90.6%)	833 (86.1%)	
All other ethnicities	164 (12.7%)	30 (9.4%)	134 (13.9%)	
**Past colonoscopy/sigmoidoscopy**				<0.001
Yes, for screening	101 (7.8%)	26 (8.1%)	75 (7.8%)	
Yes, for symptoms	260 (20.2%)	99 (30.9%)	161 (16.6%)	
No	926 (72.0%)	195 (60.9%)	731 (75.6%)	
**Past Stool test**				<0.001
Yes, for screening	475 (36.9%)	140 (43.8%)	335 (34.6%)	
Yes, for symptoms	226 (17.6%)	75 (23.4%)	151 (15.6%)	
No	586 (45.5%)	105 (32.8%)	481 (49.7%)	
**Other comorbidities (exc. diabetes)**				<0.001
0	343 (26.7%)	46 (14.4%)	297 (30.7%)	
1	382 (29.7%)	70 (21.9%)	312 (32.3%)	
2	261 (20.3%)	68 (21.3%)	193 (20.0%)	
3+	301 (23.4%)	136 (42.5%)	165 (17.1%)	
**GP visits since March 2020**				<0.001
0	370 (28.7%)	39(12.2%)	331 (34.2%)	
1	278 (21.6%)	59 (18.4%)	219 (22.6%)	
2–9	610 (47.4%)	210 (65.6%)	400 (41.4%)	
10+	29 (2.3%)	12 (3.8%)	17 (1.8%)	
**GP visits before March 2020 (per year)**				<0.001
0	445 (34.6%)	32 (10.0%)	413 (42.7%)	
1	321 (24.9%)	87 (27.2%)	234 (24.2%)	
2–9	499 (38.8%)	194 (60.6%)	305 (31.5%)	
10+	22 (1.7%)	7 (2.2%)	15 (1.6%)	

# Group removed from further analyses due to small sample size.

**Table 2 cancers-15-01668-t002:** Intended help-seeking behaviours (“definitely would” and “probably would” vs. “probably/definitely wouldn’t”) in response to potential CRC symptoms: univariable and multivariable logistic regression OR (adjusting for age, sex, ethnicity, number of comorbidities, and attributing symptom to cancer).

	Talk to Members of Your Family	Go to the Pharmacy	Contact the GP	Mention if You Saw the GP for Another Reason	Go to A&E	Look up Information Online	Wait and See What Happens	Dismiss as Something Not to Worry About	Contact a Nurse	Mention If You Saw a Nurse for Another Reason	Contact a Diabetes Specialist	Contact an Endocrinologist
	OR (95% CI)	OR (95% CI)	OR (95% CI)	OR (95% CI)	OR (95% CI)	OR (95% CI)	OR (95% CI)	OR (95% CI)	OR (95% CI)	OR (95% CI)	OR (95% CI)	OR (95% CI)
Vignette 1: Rectal bleeding												
Diabetes (unadj)												
Yes	1.01 (0.70–1.45)	1.46 (0.90–2.36)	1.61 (1.07–2.42) *	1.33 (0.73–2.42)	2.22 (1.01–4.90) *	0.50 (0.31–0.80) **	0.63 (0.44–0.90) *	0.23 (0.46–1.21)	1.86 (1.27–2.72) ***	1.03 (0.68–1.57)	5.82 (2.80–12.07) ***	1.41 (0.53–3.78)
No	1.0	1.0	1.0	1.0	1.0	1.0	1.0	1.0	1.0	1.0	1.0	1.0
Diabetes (adj)												
Yes	0.83 (0.56–1.24)	1.51 (0.87–2.54)	1.62 (1.02–2.55) *	1.32 (0.70–2.50)	2.53 (1.02–6.27) *	0.551 (0.30–0.86) *	0.60 (0.40–0.90) *	0.73 (0.43–1.24)	2.03 (1.34–3.08) ***	1.05 (0.67–1.67)	6.35 (2.83–14.2) ***	1.80 (0.61–5.33)
No	1.0	1.0	1.0	1.0	1.0	1.0	1.0	1.0	1.0	1.0	1.0	1.0
Vignette 2: Change in bowel habit												
Diabetes (unadj)												
Yes	1.21 (0.84–1.74)	1.38 (0.86–2.22)	1.60 (1.11–2.30) *	1.50 (0.98–2.30)	0.68 (0.15–3.17)	0.82 (0.54–1.23)	1.03 (0.59–1.78)	0.81 (0.56–1.16)	1.74 (1.07–2.82) *	1.30 (0.90–1.89)	7.96 (3.43–18.47) ***	0.69 (0.20–2.47)
No	1.0	1.0	1.0	1.0	1.0	1.0	1.0	1.0	1.0	1.0	1.0	1.0
Diabetes (adj)												
Yes	1.12 (0.75–1.67)	1.57 (0.94–2.63)	1.70 (1.13–2.57) *	1.74 (1.10–2.78) *	0.63 (0.12–3.45)	0.89 (0.57–1.39)	0.99 (0.55–1.81)	0.81 (0.54–1.21)	1.74 (1.03–2.96) *	1.41 (0.94–2.11)	9.48 (3.79–23.76) ***	0.49 (0.12–1.97)
No	1.0	1.0	1.0	1.0	1.0	1.0	1.0	1.0	1.0	1.0	1.0	1.0
Vignette 3: Rectal bleeding, bowel changes, and numbness and tingling in feet												
Diabetes (unadj)												
Yes	1.14 (0.79–1.63)	0.90 (0.52–1.53)	1.21 (0.72–2.04)	1.22 (0.65–2.26)	2.03 (1.05–3.91) *	0.52 (0.31–0.88) *	1.01 (0.70–1.45)	1.12 (0.68–1.85)	1.71 (1.19–2.45) **	1.17 (0.73–1.88)	4.84 (2.68–8.75) ***	2.32 (1.13–4.74) *
No	1.0	1.0	1.0	1.0	1.0	1.0	1.0	1.0	1.0	1.0	1.0	1.0
Diabetes (adj)												
Yes	1.18 (0.80–1.74)	0.91 (0.52–1.63)	1.28 (0.72–2.28)	1.31 (0.68–2.55)	2.05 (1.01–4.18) *	0.53 (0.30–0.95) *	1.07 (0.72–1.6)	1.05 (0.60–1.81)	2.00 (1.35–2.96) ***	1.44 (0.87–2.40)	5.80 (3.03–11.11 ***	2.59 (1.18–5.66) *
No	1.0	1.0	1.0	1.0	1.0	1.0	1.0	1.0	1.0	1.0	1.0	1.0

* <0.05, ** <0.01, *** <0.001.

**Table 3 cancers-15-01668-t003:** Odds ratios and 95% confidence intervals for the association between diabetes management and help-seeking among the subgroup of participants with diabetes, adjusted for age and sex (*n* = 350).

	Help-Seeking (Contacting a GP or Nurse)	*p* Value (χ^2^ Test)
Unadjusted models	OR (95% CI)	
**Annual Check**		
No	0.19 (0.09–0.40)	<0.001
Yes	1.0	

**HbA1c**		
High HbA1c *	3.08 (1.18–8.07)	0.021
Recommended HbA1c	1.0	

**Self-Management**		
Bad	0.16 (0.07–0.36)	<0.001
Average	0.63 (0.32–1.27)	0.197
Good	1.0	

** Adjusted models **		
**Annual Check**		
No	0.23 (0.10–0.50)	<0.001
Yes	1.0	

**HbA1c**		
High HbA1c	3.77 (1.36–10.50)	0.011
Recommended HbA1c	1.0	

**Self-Management**		
Bad	0.20 (0.86–0.47)	<0.001
Average	0.75 (0.37–1.54)	0.437
Good	1.0	

* Used as a marker of suboptimal control.

## Data Availability

The data that support the findings of this study are available from the corresponding author upon reasonable request.

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
