# Peer review of "The Role of Type 2 Diabetes in Patient Symptom Attribution, Help-Seeking, and Attitudes to Investigations for Colorectal Cancer Symptoms: An Online Vignette Study"

_cancers, 2023, doi:10.3390/cancers15061668_

Round 1

Reviewer 1 Report

The manuscript entitled The role of type 2 diabetes in patient symptom attribution, help-seeking, and attitudes to investigations for colorectal cancer symptoms: an online vignette study” reports a survey-based research that is aimed at examining if diabetes might influence patient symptom attribution, help-seeking and willingness to undergo investigations for possible CRC symptoms. A total of 1307 adults (340 with and 967 without diabetes) completed an online vignette survey. Participants were presented with vignettes describing new onset red-flag CRC symptoms (rectal bleeding or change in bowel habit), with or without additional symptoms of diabetic neuropathy. Survey results showed that among patients with diabetes, those not attending annual check-ups were less likely to seek help for red-flag cancer symptoms (OR=0.23; 95%Cl 0.10-0.50). Accordingly, the patients with diabetes could benefit from targeted awareness campaigns emphasizing the importance of discussing new symptoms, such as change in bowel habit, with their doctor.

The below revisions are recommended:

  1. “Participants were compensated £1.25 for completion of the survey.” Based on which criterion the compensation was determined?
  2. Section 4.5. What was the general reason of the samples who were unwilling to undergo diagnostic investigations for symptoms,  for not pursuing diagnostic investigations?
  3. Uniformity (font and size) should be mentioned throughout the manuscript including the schemes, figures, and references. The authors are encouraged to check the journal IFA.

Author Response

Thank you very much for the positive feedback on our manuscript. Please find attached the revised manuscript (with tracked changes) and our responses to reviewer's comments, indicating point-by-point how each comment was addressed.

Responses to reviewer’s comments:

1. “Participants were compensated £1.25 for completion of the survey.” Based on which criterion the compensation was determined?

Participants were paid £1.25 for completing the survey, which is based on the questionnaire taking approximately 15 minutes to complete and a £5 per hour payment. This amount was the standard set by Prolific who recruited participants from their panel and is aimed at compensating participants for the time taken to complete the survey. 

2. Section 4.5. What was the general reason of the samples who were unwilling to undergo diagnostic investigations for symptoms,  for not pursuing diagnostic investigations?

In the Results section (last sentence of section 4.5) we have mentioned the following: “In those unwilling to undergo diagnostic testing the predominant reason was embarrassment in having a stool test (19%) or a colonoscopy (26.7%) (further details in Appendix E).”

3. Uniformity (font and size) should be mentioned throughout the manuscript including the schemes, figures, and references. The authors are encouraged to check the journal IFA.

We have checked the font size and we have checked the references throughout the manuscript and have made a few minor edits (visible with track changes).

Reviewer 2 Report

The authors introduced some information about if diabetes influences patient symptom attribution, help-seeking, and willingness to undergo investigations for possible colorectal cancer symptoms. This is very important research and a good manuscript.

Author Response

Thank you very much for the positive feedback.